# Space-Constrained Scheduling Optimization Method for Minimizing the Effects of Stacking of Trades

**Han-Seong Gwak** [1,*]**, Won-Sang Shin** [1] **and Young-Jun Park** [2]

1 Construction Engineering Policy Institute of Korea, Seoul 06098, Korea; wsshin@cepik.re.kr
2 Intelligent Construction Automation Center, Kyungpook National University, Daegu 41566, Korea; py0307@knu.ac.kr
* Correspondence: hsgwak@cepik.re.kr; Tel.: +82-2-6240-4336

**Abstract:** Existing SCS (space-constrained scheduling) studies fall short of minimizing the effect of the stacking of trades that decline productivity due to an increase in resources within a physically limited work area. This article presents a space-constrained scheduling optimization (i.e., SSO) method for minimizing the stacking of trades. It imports schedule information from the project database, extracts IFC files of construction site area from the BIM model, defines the occupation density function of each activity to track the level of stacking of trades, and identifies the optimal solution (i.e., the optimal set of pairs of execution pattern alternatives and start times of activities) by implementing genetic algorithm (GA) optimization analysis. The study is of value to practitioners because SSO provides an easy-to-use computerized tool that reduces the lengthy computations relative to data processing and GAs. Test cases verify the validity of the computational method.

**Keywords:** workspace; productivity; scheduling; genetic algorithm; optimization



## 1. Introduction

Construction projects are carried out by various resources (e.g., workers, equipment, and materials), which are allocated according to the schedule plan. The resources have physical volumes and occupy a set amount of space when executing work. Existing studies named this "workspace" [1]. Construction projects are performed within a limited space and are resource-intensive; therefore, problems with workspace interference can frequently occur if the project does not have a proper schedule. Workspace interference causes a variety of problems. Generally, workspace interferences are known to affect declines in work productivity and work safety [1]. The work efficiency of labor decreases if they are not provided with sufficient workspace, and the risk of accidents increases when the workspace of construction equipment and the workspace of laborers overlap. Furthermore, if the problem goes beyond workspace interference and conflicts occur, that is, if workspaces for executing work are not provided, it becomes impossible to execute the work. Workspace conflicts inevitably lead to delayed project completion time and additional costs. Therefore, in academia, the problem of developing schedules that minimize workspace interference is named space-constrained scheduling (SCS), and many studies were conducted to address this problem [2–7]. Existing studies on SCS have had the goal of minimizing workspace interference. The general process for doing so is to (1) identify workspaces required for executing activities, (2) determine whether any workspaces are occupied simultaneously according to the sequence of activity execution, and (3) establish a plan for minimizing workspace interference.

Previous SCS studies determined whether workspace interference occurs based on the workspaces required to execute activities. This study defines such a method as an activity-oriented workspace generation and interference identification method. The activity-oriented method assumes that productivity and safety issues will not occur if activity workspaces do not overlap, even when several tasks are executed within a limited work

area. However, studies on construction productivity [8–12] have asserted that an increase in resources (e.g., workers, equipment, and materials) in a limited work area leads to declines in productivity. Previous studies called this phenomenon "stacking of trades." [8–12]. The trades are a group of skilled workers such as pipefitters, electricians, etc. Rojas [8] used statistical methods to predict declines in work productivity due to stacking of trades. The independent variable that was used in these predictions was work area density (m²/worker), whereas the dependent variable was labor productivity efficiency (percent productivity). Figure 1 shows a productivity-efficiency function for work area density in electrical work proposed by Rojas [8]. In short, productivity decreases as work area density increases. Similarly, work area density is closely related to safety issues.

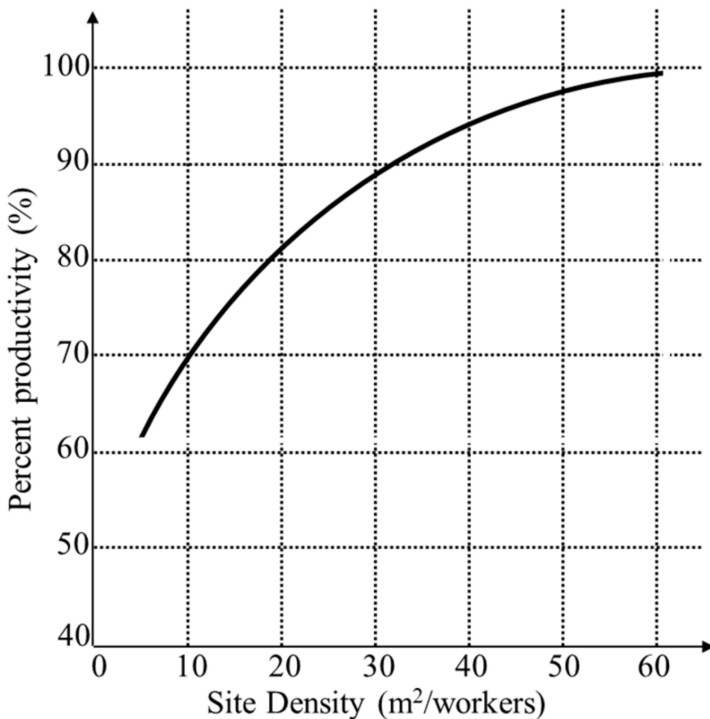

**Figure 1.** Impact of site density on labor productivity (adopted from Rojas [8]).

　　　As mentioned previously, the goal of performing SCS is to ensure work productivity and safety. Therefore, the appropriate approach for achieving the goals of SCS is an area-oriented method rather than an activity-oriented method. That is, it is necessary to have an SCS method that determines whether work areas with a certain predefined size are occupied as activities are executed, estimates the density of work areas, forecasts to what extent productivity declines when the work area is occupied simultaneously by multiple activities, and identifies a solution for minimizing workspace interference. The area-oriented SCS reduces the stacking of trades by minimizing the simultaneous occupation of the work areas that has a certain area size by multiple activities. In addition, because SCS has a quite large solution space, it is difficult to solve by intuition, and it is easy to produce poorly made plans. That is, SCS should be solved using a scientific approach, and the development of an integrated system that combines various techniques is required.

　　　This study presents an area-oriented method for establishing space-constrained schedules that minimize stacking of trades. This method was developed into a system named "Space-constrained Scheduling Optimization for Minimizing Stacking of Trades" or "SSO." In order to achieve the goals of this study, the following specific research procedures were performed. First, prior studies on SCS were analyzed to examine the necessity and uniqueness of this study, and a method development direction was established. Second, an objective function was defined considering the effect of work area interference as well

as assumptions regarding SCS that were identified through a literature review. Third, the space-constrained scheduling optimization (i.e., SSO) method was developed by combining Primavera 6 (P6), building information model (BIM), and GA. Fourth, case studies were conducted to validate the detailed application methods and performance of SSO. Finally, the research contributions and limitations were discussed.

## 2. Current State of Space-Constrained Scheduling

Most studies on SCS proposed methods for creating the workspaces needed to execute construction activities, methods for detecting workspace interference, and methods for finding strategies to resolve workspace interference. First, methods for creating the workspaces needed to execute construction activity have used modeling techniques, such as CAD, 4D CAD, and 4D BIM. Akinci et al. [2] proposed a methodology that extracts project-specific Industry Foundation Classes (IFC) from 4D CAD and creates activity workspaces based on a generic workspace ontology. Gue [13] built a hierarchical structure of workspace demand and defined the workspaces needed to execute each activity based on CAD data. In addition, changes in workspace demand according to activity progress rates were incorporated into Gue's model. Dawood and Mallasi [3] developed a 4D CAD-based space-time analysis model that identifies the workspace considering execution patterns as well as activity progress rates. Chavada et al. [4] proposed a method for real-time workspace management by combining the critical path method (CPM) and the 4D BIM model. Choi et al. [14] proposed a method that classifies workspaces as fixed workspaces or flexible workspaces based on workspace movability and identifies workspace requirements and storage space requirements for executing activities from a 4D BIM model. Getuli et al. [15] introduced a 4D BIM model for automatically creating workspaces based on the space syntax analysis.

The approaches of existing studies on workspace interference detection can be divided into three types: space interference, schedule interference, and workspace congestion. Most existing studies considered space interference. Space interference is defined as when workspaces that are needed to execute activities at the same time are physically overlapping. Therefore, it is closely related to standards for creating the workspaces needed to execute activities. However, in workspace interference, more than just physical overlapping occurs. There are also temporary overlaps between the tasks that make up the activities. This type of interference is called schedule (temporal) interference. Dawood and Mallasi [3], Moon et al. [5], and Kassem et al. [6] proposed space-constrained scheduling methods that consider space interference and schedule interference at the same time. Workspace congestion was rarely considered [5,6,9]. The workspace congestion was introduced to judge the importance of decision making when searching for workspace interference resolution methods. Gue [13] used the concepts of interference space percentage and interference duration percentage to propose the concept of workspace congestion. Moon et al. [5] introduced the schedule-workspace impact factor. Kassem et al. [6] introduced the "severity of interference" parameter to minimize schedule (temporal) interference. The severity of interference is calculated by dividing the overlapped duration by the activity duration. Dashti et al. [16] proposed a 4D BIM-based simulation model for automatic management of time–space conflicts in the pre-construction planning phase. This model enables the practitioners to detect all types of possible time–space conflicts automatically. Mirzaei et al. [17] proposed a 4D-BIM model for identifying time–space conflicts according to the labor crew's parameters movement and quantifying the impact of time–space conflicts on project performance. In addition, a study was conducted on developing an algorithm for detecting interference between resources in virtual construction site environments [18], as well as developing a GIS-based method for identifying time–space interference [19]. Existing studies on workspace interference detection have assumed that interference does not occur if activity workspaces do not overlap with each other even when several activities are executed within a limited work area. As mentioned in the introduction, a lack of studies considers the effect of the stacking of trades, in which productivity declines due to an increase in resources within a physically limited work area.

In space-constrained scheduling, it is crucial to resolve workspace interference. There are various strategies for resolving workspace interference, including changing workspace locations, workspace sizes, activity sequences, construction methods, activity execution patterns, and deferring the start times of noncritical activities [3,6,14,19]. Most of the existing studies have applied manual adjustment or rule-based heuristic methods for resolving workspace interference. Heuristic methods are easy to understand and intuitive. However, they are difficult to apply to large-scale network problems, and they have the limitation of being unable to guarantee global solutions. On the other hand, meta-heuristics, such as GA, simulation annealing (SA), and tabu search (TS), are widely used for schedule optimization because they can find quick and trustworthy near-global solutions within a large-scale solution space. However, few studies have used meta-heuristics for SCS optimization. A study by Moon et al. [5] is one of the few studies that used meta-heuristics for space-constrained scheduling. Moon et al. [5] noted that there are limitations to using manual methods to establish a reasonable workspace schedule because large-scale construction projects consist of many activities that have logical (preceding-succeeding) relationships, and they proposed a GA-based methodology for resolving workspace interference. Moon's method searches for a solution that minimizes space interference by adjusting the start times of noncritical activities within the TF (total float). That is, the solution found by GAs is an adjusted set of activity start times. Obviously, the strategy of adjusting the start times of noncritical activities may be the most preferred method of resolving workspace interference because it incurs no penalty, such as an increase in construction completion time or cost. However, changing activity execution patterns is another strategy that has no penalty [3,14,20], but it was not considered in the search for an optimal solution. Moon's method has the limitation of being unable to present a different resolution strategy when workspace interference problems cannot be resolved by adjusting activity start times.

## 3. Space-Constrained Scheduling Optimization Method

### 3.1. Method Overview

This study presents an area-oriented method for establishing schedules that minimize workspace interference while achieving minimal stacking of trades. It must be noted that this method uses different meanings for "workspace" and "work area," which could be mistaken for synonyms. Workspace implies the space that is occupied by the physical volume of resources (e.g., workers, equipment, and materials) that are allocated during the execution of activities. Therefore, workspaces differ from each activity. Work areas are spaces that divide construction sites into specific areas with certain dimensions. A work area may have been occupied by an activity's workspace. In addition, this study introduces an occupation density function to find area-oriented solutions that minimize workspace interference. This function is presented in detail later in the section describing the processes of computation. Further, to resolve workspace interference, a meta-heuristics-based optimization technique must be introduced. The optimal solution search range is focused on (1) adjustments start time of noncritical activities within the TF (total float) (2) changes in activity execution patterns that do not have penalties such as increased construction time or cost. This study adopts GA, and a GA-based optimization method for establishing schedules with minimal workspace interference is presented. The computational processes of the method are shown in Figure 2. MATLAB (ver. 2015b) was used to implement this method in the form of the software called SSO. SSO consists of five modes: (1) inputting schedule data, (2) defining occupation density function, (3) executing CPM and GA chromosome encoding, (4) defining fitness function and executing GA, (5) and outputting the near-global optimal schedules report. A detailed description of each mode is provided below.

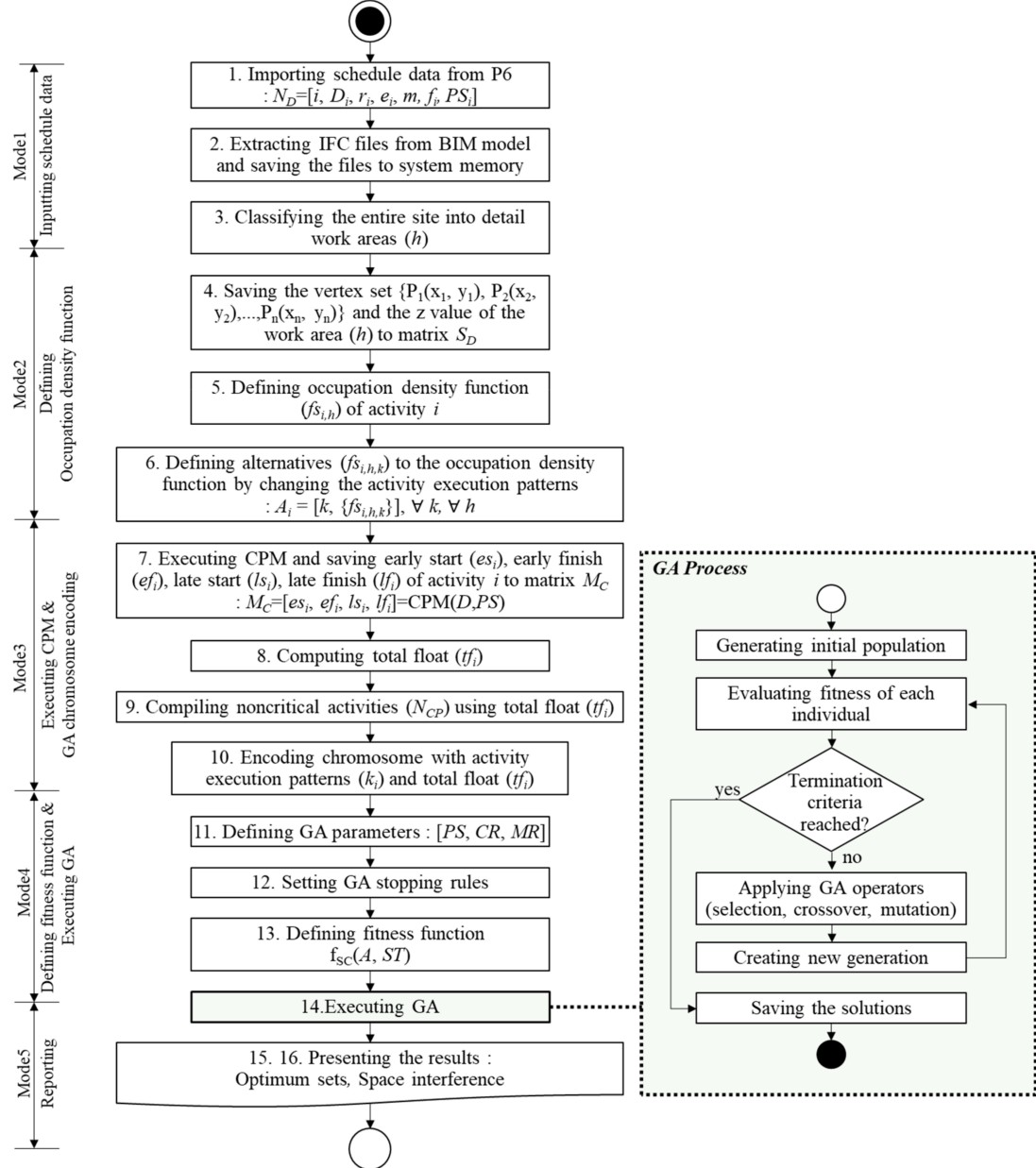

**Figure 2.** SSO algorithm.

*3.2. Importing Schedule Information from a Project Database*

In Step 1, SSO reads the schedule information (i.e., activity index ($i$), activity duration ($D_i$), the number of workers per day ($r_i$), the number of equipment per day ($e_i$), required amount of material ($m_i$), required temporary facilities ($f_i$), and predecessor ($PS_i$)) from a project database (e.g., Primavera P6), and it then saves these data into a matrix $N_D$, as shown in Equation (1). Therefore, SSO does not need to generate activity data additionally for identifying optimal space-constrained schedules by incorporating with Primavera P6, which is a commercial SW for schedule management.

$$N_D = [i, \ D_i, r_i, m_i, f_i, PS_i], \ i = 1 : I \tag{1}$$

Next, SSO acquires all construction sites areas that are required to execute the project (Step 2). For this, the Industry Foundation Classes (IFC) file extracted from the BIM model is stored in the system. The IFC file is a text file contacting entity instances according to

the IFC EXPRESS schema. The location information of the work areas is managed by the IfcProduct entity. The IfcProduct entity has a basic data structure that expresses the location and shape of its sub-entity IfcBuildingElement. The work area planes are made of lines that have (x, y) values, and the plane levels are classified by z values.

### 3.3. Defining the Occupation Density Function

The user with expertise in construction planning and management divides the construction site into specific work areas considering the characteristics of the workspace (Step 3). For example, rooms that are closed on all sides, transporting passages, and material loading spaces, etc., can be classified into specific work areas. Certainly, it is simply possible to set up the specific work areas by dividing the site into grid unit planes. However, it should be noted that the specific work areas (*h*) that are defined in this step are the standard for defining the occupation density function in Step 5. If the construction site is divided into too many specific work areas, it may affect the computation speed of finding the optimal solution.

A defined specific work area (*h*) can be converted into a polygonal plane that is made up of several vertices. SSO stores the vertex coordinates set {$P_1(x_1, y_1)$, $P_2(x_2, y_2)$,...,$P_n(x_n, y_n)$} and z value of the specific work area (*h*) in the matrix $S_D$ (Equation (2)).

$$S_D = [(x_{h,n}, y_{h,n}), z_h], \ n = 1 : N, \ h = 1 : H \tag{2}$$

Next, it is required to determine a function for the density at which a certain work area is occupied due to the workspace needed to execute an activity (*i*) in Step 5. In this study, this is named the occupation density function (*fs*). The occupation density of the work area may change according to the progress rate of the activity. The occupation density may be low during the planning stage and high during the end stage. The opposite scenario may also be true. In addition, the work areas that are occupied may vary according to the rate of progress. For example, in the case of masonry work, transport routes are necessary for the initial process, but the routes are no longer required for stacking bricks in the subsequent process. In this way, occupied work areas and occupation density vary with characteristics of activity and rate of progress. SSO identifies the occupation density function (*$fs_{i,h}$*), where the *x*-axis is the rate of progress (*p*, *p* = 0:100) and the y axis is the occupation density (*w*, *w* = 0:1) for the specific work area (*h*) of each activity (*i*), as shown in Equation (3). An occupation density of 1 means the maximum allowable value at which work execution is possible in the work area.

$$fs_{i,h}(p) = \frac{r_{i,h}(p) + e_{i,h}(p) + m_{i,h}(p) + f_{i,h}(p)}{s_h} \tag{3}$$

where *h* is the index of specific work areas defined in step 3; $r_{i,h}(p)$ is the area occupied by the workers in the work area *h* when the rate of progress for activity *i* is *p*; $e_{i,h}(p)$ is the area occupied by equipment in work area *h* when the rate of progress for activity *i* is *p*; $m_{i,h}(p)$ is the area occupied by material in work area *h* when the rate of progress for activity *i* is *p*; $f_{i,h}(p)$ is the area occupied by temporary facilities in work area *h* when the rate of progress for activity *i* is *p*; and $s_h$ is the area size of work area *h*.

$r_{i,h}(p)$, $e_{i,h}(p)$ and $m_{i,h}(p)$ are calculated as unit size × number of inputs. The unit size for each variable is obtained from the material DB, and the number of inputs is obtained from the matrix ($N_D$) defined in Step 1. $s_h$ is calculated using the polygon-area function (Equation (4)). The vertex coordinates of work area (*h*), which are variables used in the $s_h$ calculation, are obtained from the matrix $S_D$ stored in Step 4.

$$s_h = polygonarea(x_n, y_n, n) = \frac{1}{2} \left| \sum_{n=1}^{N} (x_n + x_{n+1})(y_n - y_{n+1}) \right| \tag{4}$$

Figure 3 is a simple example of explaining the concept of the occupation density function. It is assumed that the work area (*h*) was divided into four areas (A, B, C, and D) in Step 3. Activity *i* is executed by occupying the A work area until the rate of progress is 40%, and the area occupation density is 0.4. After the progress rate of 40%, work area B is occupied at 0.2 density. After 80%, work area B is occupied at a density of 0.2, and simultaneously, work area D is occupied. However, the occupation density of work area D increases at a constant rate from 0.4 to 0.6. Work area C is not occupied, while activity *i* is executed. Occupation density functions can be defined in various ways according to the characteristics and work areas of the activities.

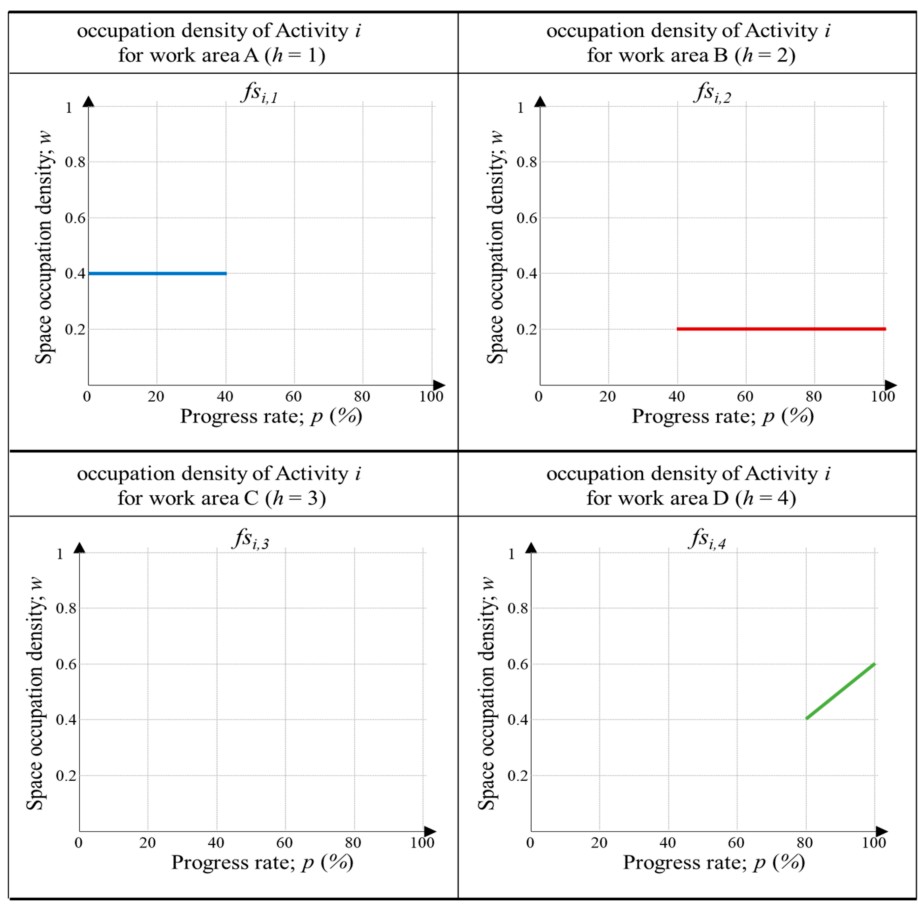

**Figure 3.** An example of the occupation density function.

In order to identify the occupation density function of work areas, the user must define the work areas occupied according to the progress rate of the activity, which requires expert judgment. However, this process can be automated by integrating 4D-BIM, which includes execution schedule and spatial information for activities, into SSO.

The definition of execution patterns is the direction in which an activity is to be executed [3]. The strategy of changing execution patterns in SCS has become widely used to minimize workspace interference without penalties such as increases in time and cost [3,14,20]. In this study, the changing activity execution patterns method is used to create alternatives for occupation density function (*fs*). In other words, when different activity execution patterns are used; i.e., when the direction of work progress is different, the variables in Equation (3), such as $r_{i,h}$, $e_{i,h}$, $m_{i,h}$, and $f_{i,h}$, vary, and the work areas (*h*) that are occupied according to the rate of progress (*p*) differ. As a result, a different occupation density function ($fs_{i,h}$) is defined. Alternatives for the occupation density function (*fs*) of

each work area ($h$) obtained by changing the execution pattern of activity ($i$) are stored in a matrix $A_i$ (Step 6), as shown in Equation (5).

$$A_i = \begin{bmatrix} 1 & \{fs_{i,h,1}\} \\ 2 & \{fs_{i,h,2}\} \\ \vdots & \vdots \\ k & \{fs_{i,h,k}\} \\ \vdots & \vdots \\ K_i & \{fs_{i,k,Ki}\} \end{bmatrix}, \forall h \tag{5}$$

where $K_i$ is the number of alternatives for the occupation density function of activity $I$, and $k$ is the index of an alternative.

### 3.4. Executing CPM and GA Chromosome Encoding

SSO obtains the activity duration ($D_i$) and predecessor ($PS_i$) information from the $N_D$ and executes the CPM algorithm using the given $D_i$ and $PS_i$ (Step 6), as shown in Equation (6). The computed schedule data (i.e., early start times ($es_i$), early finish times ($ef_i$), late start times ($ls_i$), and late finish times ($lf_i$) of each activity ($i$)) calculated by the CPM are stored in a matrix ($M_C$) (Equation (7)). The total float ($tf_i$) of each activity ($i$) is calculated as the difference between the late start time ($ls_i$) and early start time ($es_i$) (Equation (8)). Afterward, SSO compiles activities with total floats ($tf$) greater than zero (i.e., $tf_i > 0$) as noncritical activities ($N_{CP}$) and sets $j$ as an index of $N_{CP}$ (Step 9).

$$[es_i, ef_i, ls_i, lf_i] = CPM(D_i, PS_i), \ i = 1 : I \tag{6}$$

$$M_C = [es_i, ef_i, ls_i, lf_i], \ i = 1 : I \tag{7}$$

$$tf_i = ls_i - es_i, \ \forall i \tag{8}$$

SSO uses two strategies for minimizing space interference. The first is selecting an alternative for the execution patterns of each activity, and the second is deferring the start times ($st_i$) of noncritical activities ($N_{CP}$). GA chromosomes are composed of a set of genes. In SSO, the GA chromosome is composed of a pair of the index ($k_i$) of the alternatives of execution patterns for activity ($i$) and the number of deferred days ($sd_i$) of the start time for the activity ($i$) (Figure 4). The index ($k_i$) of alternatives of execution pattern is smaller than or equal to the number of alternatives ($K_i$) of occupation density function for activity $i$ found in Step 6 ($k_i \leq K_i$). The number of deferred days ($sd_i$) is always zero if activity $i$ is a critical activity ($sd_i = 0, i \not\subset N_{CP}$), and it is smaller than or equal to the total float ($tf_j$) found in Step 9 if activity $i$ is a noncritical activity ($sd_i \leq tf_j, I \subset N_{CP}$). The serial number of a gene pair is the same as the index ($i$) of activity. The search range of the optimal solution ($S_S$) is the number of combinations of the execution pattern alternatives and the range of deferrable days of activities (Equation (9)).

$$S_S = \prod_i (k_i \times sd_i) \tag{9}$$

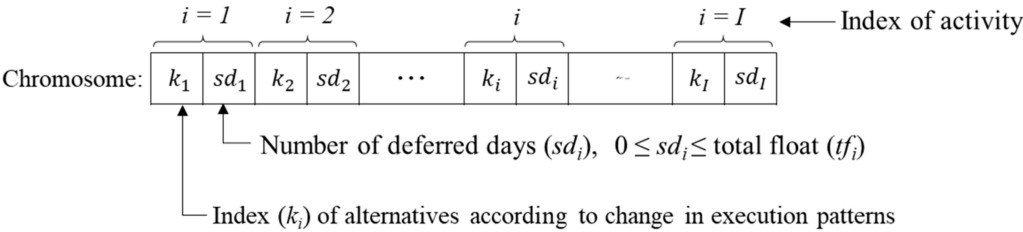

**Figure 4.** Encoding a chromosome.

### 3.5. Defining the Objective Function and Executing GA

Meta-heuristics such as GA, SA, and TS are effective in solving the problem in this study. All three meta-heuristics have the ability to find a better solution than the manual process. Several studies on the performance comparison of GA, SA, and TS were conducted [21–23]. Lidbe et al. [21] show that TS gives better calibration results compared to the GA and SA. On the other hand, Said et al.'s [23] study found that GA has a better solution quality than SA and TS for solving QAP (Quadratic Assignment Problem) optimization problems, but TS has a faster execution time than the others. This suggests that there may be differences in the performance of GA, SA, and TS depending on the type of problem. Therefore, this study adopts GA, which is the most diverse and widely used in the construction field, to solve the SCS problem.

The GA parameters, fitness function, and GA operation termination rules used to find solutions are provided in detail. First, the user sets GA parameters (i.e., population size (*PS*), crossover rate (*CR*), mutation rate (*MR*)) to execute GA operations (Step 11). Then, SSO sets the GA operation termination rules. For the termination rules, we used the four termination criteria provided by Global Optimization ToolBox of MATLAB (ver. 2015b) developed by MathWorks, USA: (1) Generation: the operation is terminated if the maximum number of generations is reached; (2) StallGenLimit: the operation is terminated if there is no improvement in the objective function during a specific number of generations (stall generation); (3) StallTimeLimit: the operation is terminated if there is no improvement in the objective function for a certain period; (4) TimeLimit: the operation is terminated if the maximum computation time is exceeded. Tolfun ($=1 \times 10^{-6}$), which is set as a default value by the GA solver of MATLAB, is used to determine whether the objective function has improved.

SSO sets the objective function of the GA (Step 13). The objective function $f_{SC}$(A, ST) calculates the level of space interference given the set of alternatives of activity execution pattern A($=\{k_1,...,k_I\}$) and the set of deferred start times ST($=\{sd_1,...,sd_I\}$) using Equation (10). Space interference occurs when the work area (*h*) is occupied by multiple activities at the same project time point (*t*). Therefore, space interference minimization is to plan that the work is not executed in the same area simultaneously. When it is unavoidable for multiple activities to be performed simultaneously in the same work area due to the nature of the project, it is necessary to have a plan that arranges activities that have a small space interference effect of being in the same area simultaneously. Equation (10) does not add the occupation density if the work area (*h*) is not occupied ($a_{t,h} = 0$), even if several activities are performed simultaneously at a certain project time (*t*). If several activities are performed in the same area at the same time ($a_{t,h} = 1$), it means that space interference has occurred, and the corresponding amount of occupation density is added. If the sum of the area occupation density at time *t* is greater than 1, a penalty (*p*) is added. This is because the maximum allowed density of work that can be executed in the work area has been exceeded, as mentioned in Step 5.

$$f_{SC}(A, ST) = \sum_t \sum_h ((fs_{i,h,k}(p_{i,t}) \times a_{t,h}) + P_{t,h})$$
$$s.t. \ p_{t,i} = \frac{t - st_i}{D_i} \times 100$$
$$a_{t,h} = \begin{cases} 1, & n(i|fs_{i,h,k}(p_{i,t}) > 0, \ \forall i) > 1 \\ 0, & otherwise \end{cases} \quad (10)$$
$$P_{t,h} = \begin{cases} p, & if \ \sum_i (fs_{i,h,k}(p_{i,t}) \times a_{t,h}) > 1 \\ 0, & otherwise \end{cases}$$

where $fs_{i,h,k}$ is the occupation density function for work area *h* when the execution pattern alternative for activity *i* is *k*; $p_{i,t}$ is the progress rate of activity *i* at time *t*; $a_{t,h}$ is a binary number that determines whether work area *h* occupied simultaneously by multiple activities at time t; $P_{t,h}$ is the penalty (*p*) when the sum of the occupation density in work area *h* at time *t* is greater than 1.

By using a simple example with two activities, P and Q, and two work areas, A and B, Figure 5 shows the procedure for calculating the level of space interference and the procedure in which the level of space interference changes with the application of alternatives for execution pattern and deferred start time of activity P.

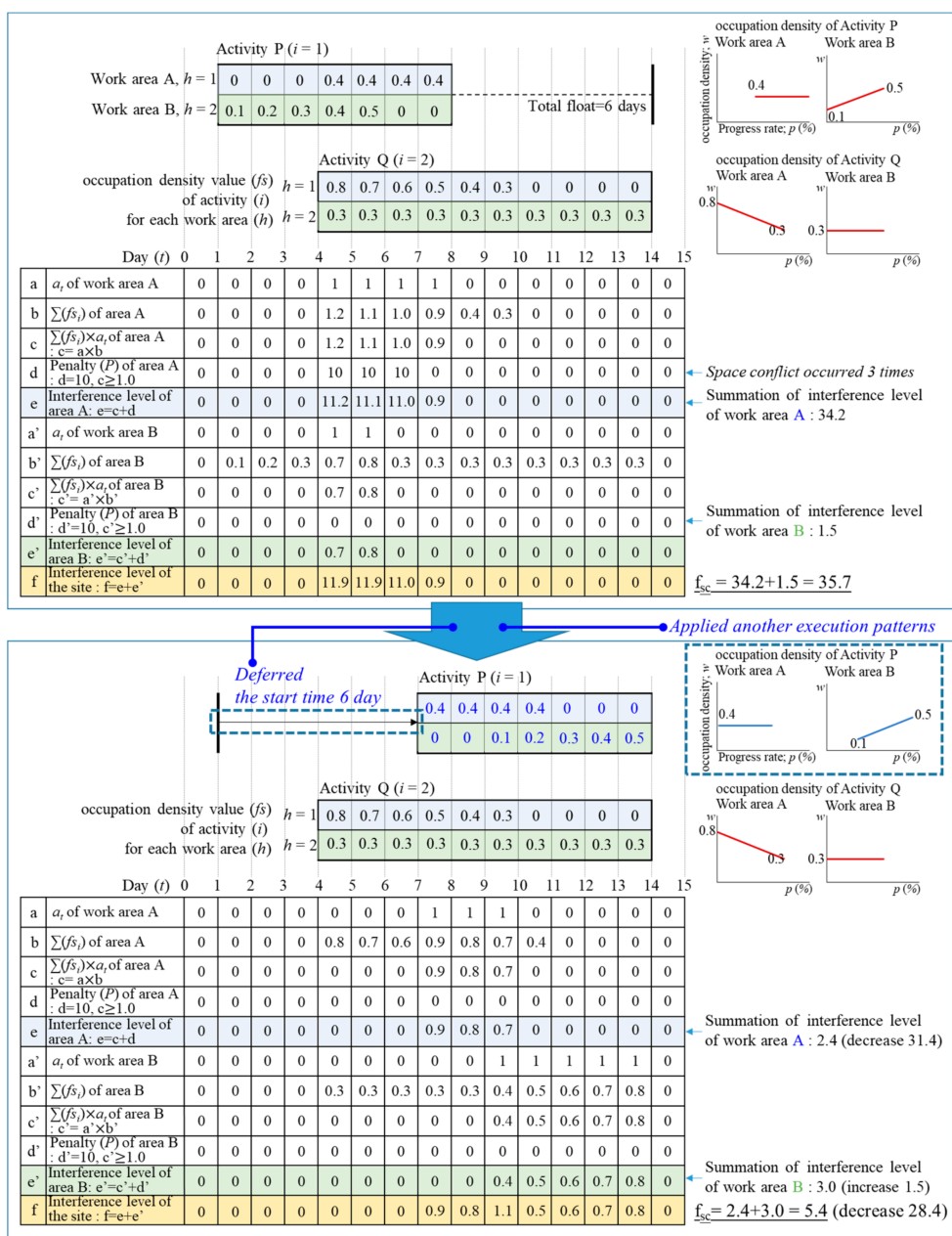

**Figure 5.** A simple example shows the procedure for calculating and minimizing the level of space interference.

As the optimal schedule minimizes the level of space interference, the fitness function is set as the inverse of the level of space interference. Afterward, SSO executes the GA (Step 14). The GA generates the initial population based on *PS*, which is a previously defined population size, and gradually improves the chromosomes via numerous evolutionary generations to converge to the near-global optimal solution. The superior and inferior chromosomes of each generation are distinguished by the fitness function, and the superior chromosomes are selected with high probability. Eventually, the inferior chromosomes are removed as the generation progresses, and the process of mutating (mutation) or exchanging (crossover) the genetic information between the superior chromosomes is repeated. The GA iterations stop as soon as any of the stopping rules are satisfied.

### 3.6. Output Near-Global Optimal Schedule

The fitness function identifies the optimal set of pairs of execution pattern alternatives and deferred start times of activities that minimize space interference. After completing the optimization operation of the GA, SSO provides the user with visual information, such as (1) the schedule with the application of execution pattern alternative for each activity, (2) the schedule with the application of deferred start times of activities, (3) the times when space interference occurs, and (4) the level of space interference.

## 4. Method Verification

### 4.1. Verifying the Effectiveness of SSO

The activity-on-node network consisting of 13 activities shown in Figure 6 was used to illustrate the procedure described in the preceding section and allows us to validate each step of the SSO method in detail. The case study is a network for electrical and mechanical work on a site with the floor plan as shown in Figure 7. The schedule information is stored in Table 1 with the activity index ($i$), activity duration ($D_i$), the number of workers per day ($r_i$) assigned to an activity, the number of equipment per day ($e_i$), required amount of material ($m_i$), required temporary facilities ($f_i$), and predecessor ($PS_i$), etc. The case site was divided into three specific work areas ($h$), and their area size were 51.3 m$^2$, 42.1 m$^2$, and 98.6 m$^2$. The occupation density function ($fs$) for each activity was found by considering the resources needed to execute the activity and the characteristics (e.g., location and area size) of specific work areas (Table 2). The number of alternatives for occupation density function ($fs$) obtained by changing the execution pattern of activity ($i$) were [3, 2, 2, 1, 4, 2, 1, 2, 2, 1, 2, 2, 3]. The CPM computes the critical path and total project completion time as A- > D- > L and 66 days, respectively. The non-critical activities ($N_{CP}$) are B, C, E, F, G, H, I, J, K, and M. The total floats ($tf_i$) of activities were [0, 8, 17, 0, 9, 8, 9, 8, 17, 27, 8, 0, 27]. Therefore, based on Equation (9), the size of the optimal solution search range in this test case was

$$7.68 \times 10^{14} = (3 \times 1) \times (2 \times 9) \times (2 \times 18) \times (1 \times 1) \times (4 \times 10) \times (2 \times 9) \times (1 \times 10) \times (2 \times 9) \times (2 \times 18) \times (1 \times 28) \times (2 \times 9) \times (2 \times 1) \times (3 \times 28).$$

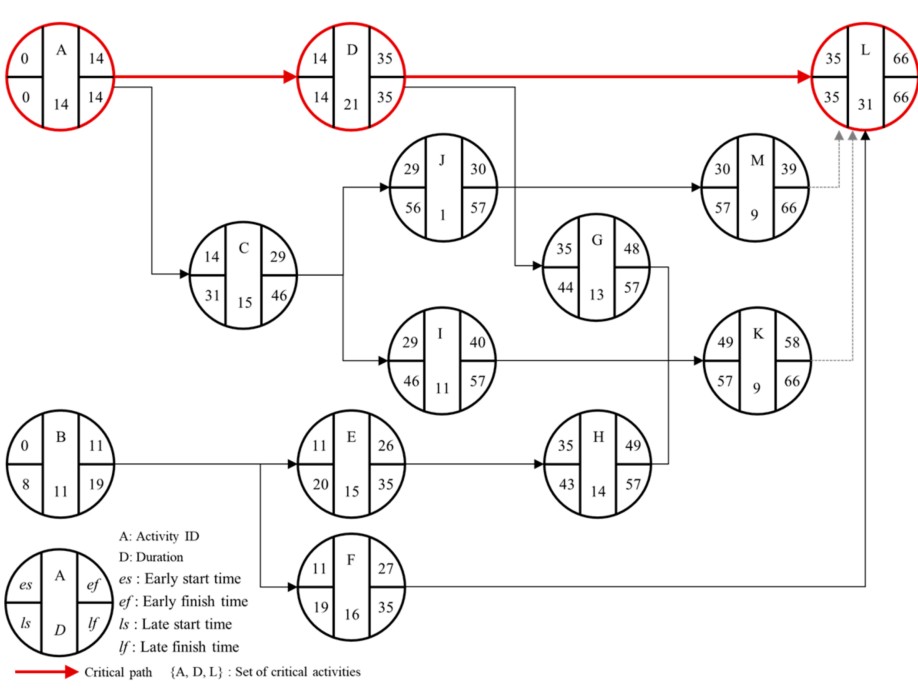

**Figure 6.** Network for the case study.

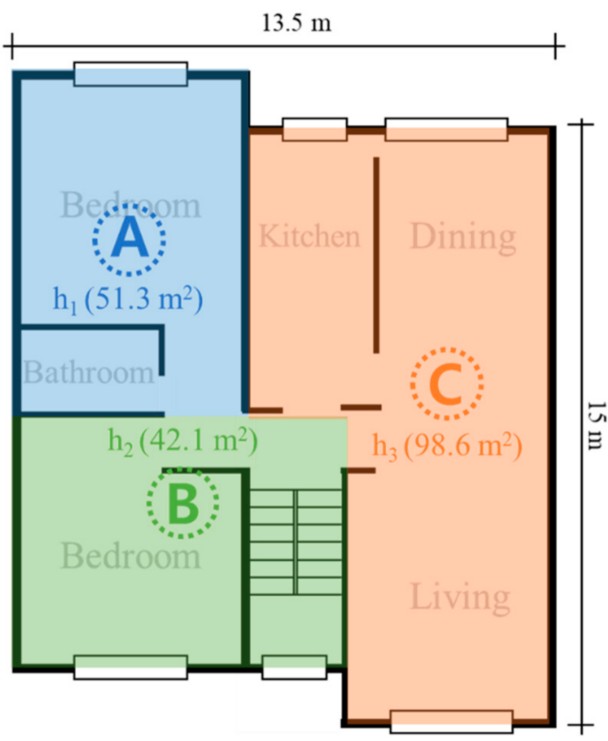

**Figure 7.** Floor plan, Specific work areas (*h*).

**Table 1.** Schedule information of the case study.

| Activity ID | Activity Index (*i*) | Predecessors (*PS_i*) | Duration (*D_i*) | Workers Num. (Unit Size) (*r_i*) | Equip. Num. (Unit Size) (*e_i*) | Material Qty. (Unit Size) (*m_i*) | Temp. Facility Qty. (unit size) (*f_i*) | Total Float (*tf_i*) | Alt. Num. (*k_i*) |
|---|---|---|---|---|---|---|---|---|---|
| A | 1 | - | 14 | 5 (4 m$^2$) | 1 (10 m$^2$) | - | 2 (8 m$^2$) | 0 | 3 |
| B | 2 | - | 11 | 3 (4 m$^2$) | 1 (10 m$^2$) | 23 (1 m$^2$) | - | 8 | 2 |
| C | 3 | A | 15 | 2 (4 m$^2$) | 1 (15 m$^2$) | 15 (2 m$^2$) | 1 (10 m$^2$) | 17 | 2 |
| D | 4 | A | 21 | 2 (4 m$^2$) | - | - | 2 (10 m$^2$) | 0 | 1 |
| E | 5 | B | 15 | 5 (4 m$^2$) | 1 (15 m$^2$) | - | - | 9 | 4 |
| F | 6 | B | 16 | 3 (4 m$^2$) | - | 15 (2 m$^2$) | - | 8 | 2 |
| G | 7 | D,E | 13 | 3 (4 m$^2$) | - | - | - | 9 | 1 |
| H | 8 | D,E,F | 14 | 5 (4 m$^2$) | - | 5 (1 m$^2$) | 1 (10 m$^2$) | 8 | 2 |
| I | 9 | C | 11 | 3 (4 m$^2$) | - | 8 (1 m$^2$) | 2 (7 m$^2$) | 17 | 2 |
| J | 10 | C | 1 | 1 (4 m$^2$) | 2 (15 m$^2$) | 20 (1 m$^2$) | 1 (5 m$^2$) | 27 | 1 |
| K | 11 | G,H,I | 9 | 3 (4 m$^2$) | - | 5 (4 m$^2$) | 1 (8 m$^2$) | 8 | 2 |
| L | 12 | D,E,F | 31 | 3 (4 m$^2$) | 1 (15 m$^2$) | 15 (2 m$^2$) | - | 0 | 2 |
| M | 13 | J | 9 | 2 (4 m$^2$) | 3 (5 m$^2$) | 8 (1 m$^2$) | - | 27 | 3 |

**Table 2.** Occupation density function alternatives obtained by changing the execution pattern of activity.

| Activity ID (Index) | Index of Alt. (*k_i*) | Occupation Density Function (*fs_{i,h,k}(p)*, *p*: Progress Rate) | | |
|---|---|---|---|---|
| | | Work Area A (*h* = 1) | Work Area B (*h* = 2) | Work Area C (*h* = 3) |
| A (*i* = 1) | 1 | $fs_{1,1,1}(p) = 0.4, 0 < p \leq 0.7$ | $fs_{1,2,1}(p) = 0.33p - 0.03, 0.7 < p \leq 1$ | $fs_{1,3,1}(p) = 0.3, 0 < p \leq 0.7$ |
| | 2 | $fs_{1,1,2}(p) = 0.33p - 0.03, 0.7 < p \leq 1$ | $fs_{1,2,2}(p) = 0.3, 0 < p \leq 0.7$ | $fs_{1,3,2}(p) = 0.3, 0 < p \leq 0.7$ |
| | 3 | $fs_{1,1,3}(p) = -0.2p + 0.3, 0.5 < p \leq 1$ | $fs_{1,2,3}(p) = 0.1, 0 < p \leq 1$ | - |
| B (*i* = 2) | 1 | $fs_{2,1,1}(p) = 0.2p + 0.1, 0 < p \leq 0.5$ | $fs_{2,2,1}(p) = -0.41(p - 0.3)^2 + 0.4, 0 < p \leq 0.5$ | - |
| | 2 | $fs_{2,1,2}(p) = -0.41(p - 0.3)^2 + 0.4, 0 < p \leq 0.5$ | $fs_{2,1,1}(p) = 0.2p + 0.1, 0 < p \leq 0.5$ | - |

**Table 2.** *Cont.*

| Activity ID (Index) | Index of Alt. ($k_i$) | Occupation Density Function ($fs_{i,h,k}(p)$, $p$: Progress Rate) | | |
| --- | --- | --- | --- | --- |
| | | Work Area A ($h = 1$) | Work Area B ($h = 2$) | Work Area C ($h = 3$) |
| C ($i = 3$) | 1 | $fs_{3,1,1}(p) = 0.3, 0 < p \leq 0.5$ | $fs_{3,2,1}(p) = 0.8p, 0 < p \leq 0.5$ | $fs_{3,3,1}(p) = 1/2.5\log(p + 1) + 0.2, 0 < p \leq 0.8$ |
| | 2 | $fs_{3,1,2}(p) = 0.4p + 0.1, 0 < p \leq 0.5$ | $fs_{3,2,2}(p) = 0.1, 0.5 < p \leq 1$ | $fs_{3,3,2}(p) = 1/0.8\log(p + 1) + 0.5, 0 < p \leq 0.8$ |
| D ($i = 4$) | 1 | - | $fs_{4,2,1}(p) = 1.1p - 0.8, 0 < p \leq 0.7$ | $fs_{4,3,1}(p) = 0.2, 0.3 < p \leq 0.7$ |
| E ($i = 5$) | 1 | $fs_{5,1,1}(p) = 0.4, 0 < p \leq 0.4$ | $fs_{5,2,1}(p) = 0.33p - 0.03, 0.2 < p \leq 0.8$ | $fs_{5,3,1}(p) = -2.5(p - 0.8)^2 + 0.4, 0.8 < p \leq 1$ |
| | 2 | $fs_{5,1,2}(p) = 0.33p - 0.03, 0.2 < p \leq 0.8$ | $fs_{5,2,2}(p) = 0.2, 0 < p \leq 0.4$ | $fs_{5,3,2}(p) = -5(p - 0.8)^2 + 0.7, 0.8 < p \leq 1$ |
| | 3 | $fs_{5,1,3}(p) = 0.2, 0 < p \leq 0.3$ | $fs_{5,2,3}(p) = 0.14p + 0.16, 0.3 < p \leq 1$ | $fs_{5,3,3}(p) = 1/0.4\log(p + 0.8) + 0.1, 0.2 < p \leq 0.4$ |
| | 4 | $fs_{5,1,4}(p) = 1/0.4\log(p + 0.8) + 0.1, 0.2 < p \leq 0.4$ | $fs_{5,2,4}(p) = 0.14p + 0.16, 0.3 < p \leq 1$ | $fs_{5,4,4}(p) = 0.2, 0 < p \leq 0.3$ |
| F ($i = 6$) | 1 | $fs_{6,1,1}(p) = 1/0.56\log(p + 0.8) + 0.2, 0.2 < p \leq 0.5$ | $fs_{6,2,1}(p) = 0.2, 0 < p \leq 0.5$ | $fs_{6,3,1}(p) = 0.7^{(p - 0.4)} - 0.7, 0.4 < p \leq 1$ |
| | 2 | $fs_{6,1,2}(p) = 0.2, 0 < p \leq 0.5$ | $fs_{6,2,2}(p) = 1/0.5\log(p + 0.8) + 0.2, 0.2 < p \leq 0.5$ | $fs_{6,3,2}(p) = 0.7^{(p - 0.4)} - 0.6, 0.4 < p \leq 1$ |
| G ($i = 7$) | 1 | $fs_{7,1,1}(p) = 0.2, 0 < p \leq 1$ | - | - |
| H ($i = 8$) | 1 | - | $fs_{8,2,1}(p) = -0.5(p - 0.4)^2 + 0.6, 0.4 < p \leq 1$ | $fs_{8,3,1}(p) = 0.1, 0 < p \leq 0.5$ |
| | 2 | $fs_{8,1,2}(p) = 0.1, 0 < p \leq 0.5$ | $fs_{8,2,2}(p) = 0.2, 0.3 < p \leq 0.7$ | $fs_{8,3,2}(p) = -2.2(p - 0.7)^2 + 0.6, 0.7 < p \leq 1$ |
| I ($i = 9$) | 1 | $fs_{9,1,1}(p) = 0.14p + 0.4, 0 < p \leq 0.7$ | $fs_{9,2,1}(p) = -0.2p + 0.3, 0.5 < p \leq 1$ | - |
| | 2 | $fs_{9,1,2}(p) = -0.2p + 0.3, 0.5 < p \leq 1$ | $fs_{9,2,2}(p) = 1.4p + 0.2, 0 < p \leq 0.7$ | - |
| J ($i = 10$) | 1 | - | $fs_{10,2,1}(p) = 0.5, 0 < p \leq 1$ | $fs_{10,3,1}(p) = 0.4(p - 0.3)^2 + 0.2, 0.3 < p \leq 1$ |
| K ($i = 11$) | 1 | - | - | $fs_{11,3,1}(p) = -0.2p + 0.4, 0 < p \leq 1$ |
| L ($i = 12$) | 1 | $fs_{12,1,1}(p) = 0.2p + 0.2, 0 < p \leq 1$ | $fs_{12,2,1}(p) = 0.8^{(p - 0.4)} - 0.8, 0.4 < p \leq 1$ | $fs_{12,3,1}(p) = 1/0.4\log(p + 0.8) + 0.1, 0.2 < p \leq 0.4$ |
| | 2 | $fs_{12,1,2}(p) = 0.8^{(p - 0.4)} - 0.8, 0.4 < p \leq 1$ | $fs_{12,2,2}(p) = 0.1p + 0.1, 0 < p \leq 1$ | $fs_{12,3,2}(p) = 1/0.4\log(p + 0.8) + 0.1, 0.2 < p \leq 0.4$ |
| M ($i = 13$) | 1 | $fs_{13,1,1}(p) = 0.4, 0 < p \leq 0.3$ | $fs_{13,2,1}(p) = -0.3p + 0.6, 0 < p \leq 1$ | - |
| | 2 | $fs_{13,1,2}(p) = 0.5, 0 < p \leq 1$ | $fs_{13,2,2}(p) = 0.4(p - 0.3)^2 + 0.2, 0.3 < p \leq 1$ | - |
| | 3 | $fs_{13,1,3}(p) = -0.3p + 0.4, 0 < p \leq 1$ | $fs_{13,2,2}(p) = 0.2, 0 < p \leq 0.3$ | - |

The space interference level ($f_{SC}$) of the original schedule, which is computed by commencing each activity at its original EST ($es_i$), was 47.48. The space interference level of each specific work area was 8.85, 16.35, and 22.28. In the case of exceeding the occupation allowance of the specific work areas ($\sum fs > 1$) was found to occur three times (at 23, 24, and 30 days). The space interference level ($f_{SC}$) of the LST ($ls_i$) schedule was calculated to be 66.58 ($h_1 = 29.36$, $h_2 = 32.88$, $h_3 = 4.34$). The allowed occupation density of specific work areas was exceeded four times (at 46, 54, 56, and 57 days). Because early start time (EST) and late start time (LST) schedules both exceeded the allowed value for occupation density, it was predicted that there would be inevitable construction delays and cost increases. Therefore, it is necessary to establish a schedule that minimizes space interference by performing optimization.

The GA parameters were initialized with [$PS$ = 400, $CR$ = 0.4, $MR$ = 0.05, StallGenLimit = 200 times, TimeLimit = Inf, StallTimeLimit = Inf] to execute the GA optimization operation. The GA operation stopped as the allowable cumulative change of the objective function reached $1 \times 10^{-6}$ after undergoing 124 generations.

SSO found [(3,0), (1,0), (2,0), (1,0), (1,4), (2,8), (1,1) (2,0), (1,13), (1,0), (1,2), (1,0), (3,1)] as the near-global solution (i.e., the set of pairs of execution pattern alternatives and deferred start times of activities) that minimizes space interference without exceeding the occupation allowance of each specific work area (Table 3). These results were expressed using the chromosomes designed for the GA operation. Converting these results into alternatives for activity execution patterns ($k_i$) yields [3,1,2,1,1,2,1,2,1,1,1,1,3], and the number of deferred days of activities start time ($sd_i$) is [0,0,0,0,4,8,1,0,13,0,2,0,1]. Figure 8 shows a diagram of the obtained results. When the network was executed using the found solution, the space interference level ($f_{SC}$) was calculated to be 17.79. This was a space interference level reduction of 30.29 compared with the EST schedule and 49.39 compared with the LST schedule. The space interference levels for each specific work area were 8.62, 5.77, and 3.40. There were no cases in which the occupation allowance of specific work areas was exceeded ($\sum fs > 1$). Therefore, there is no need for concern regarding construction delays and cost increases. It was confirmed that SSO finds a solution that minimizes space interference without exceeding the allowed occupation density of each work area.

**Table 3.** Experimental results.

| Activity ID | A | B | C | D | E | F | G | H | I | J | K | L | M |
|---|---|---|---|---|---|---|---|---|---|---|---|---|---|
| $i$ | 1 | 2 | 3 | 4 | 5 | 6 | 7 | 8 | 9 | 10 | 11 | 12 | 13 |
| Solution ($k_i$, $sd_i$) | (3, 0) | (1, 0) | (2, 0) | (1, 0) | (1, 4) | (2, 8) | (1, 1) | (2, 0) | (1, 13) | (1, 0) | (1, 2) | (1, 0) | (3, 1) |
| Execution pattern Alt. ($k_i$) | 3 | 1 | 2 | 1 | 1 | 2 | 1 | 2 | 1 | 1 | 1 | 1 | 3 |
| Deferred start time ($sd_i$) | 0 | 0 | 0 | 0 | 4 | 8 | 1 | 0 | 13 | 0 | 2 | 0 | 1 |
| Exceeded occupation allowance | Does not occur | | | | | | | | | | | | |
| Space interference level ($f_{SC}$) | 17.79 (8.62 on A work area; 5.77 on B work area; 3.40 on C work area) | | | | | | | | | | | | |

The performance of GA can be changed depending on the values chosen for the GA parameters. Therefore, performing GA once with only one set of GA parameters cannot guarantee the reliability of the obtained results. In order to improve reliability, the sensitivity analysis was performed as follows. This analysis starts by setting the minimum, incremental, and maximum value of each GA parameter (i.e., population size ($PS$) = [$PSmin$:$IPS$:$PSmax$] = 300:20:500), crossover rate (($CR$) = [$CRmin$:$ICR$:$CRmax$] = 0.4:0.1:0.8), and mutation rate (($MR$) = [$MRmin$:$IMR$:$MRmax$] = [0.05:0.01:0.08]). In this example, there are 11 population sizes (300, 320, 340, 360, 380, 400, 420, 440, 460, 480, 500), 5 crossover rates (0.4, 0.5, 0.6, 0.7, 0.8), and 4 mutation rates (0.05, 0.06, 0.07, 0.08). Therefore, a total of 220 (=11 × 5 × 4) GA experiments are conducted, and the experimental results are saved along with the space interference level ($f_{SC}$) and the number of evolved generations ($g$). The results are shown in Figure 9. The results of sensitivity analysis present that the interference level ($f_{SC}$) is the minimum of 17.79, and the maximum is 24.35. The minimum $fsc$ was searched for eight parameter combinations, [320-0.5-0.05], [340-0.7-0.07], [400-0.4-0.05], [400-0.6-0.07], [400-0.8-0.06], [420-0.7-0.08], [460-0.5-0.06], and [500-0.4-0.07], respectively. It was confirmed that the obtained results with the initial set of GA parameters (i.e., $PS$ = 400, $CR$ = 0.4, $MR$ = 0.05) were the optimal solution.

### 4.2. Verifying the Outperformance of SSO for Handling a Large Network

Another real-life plant construction project network with 134 activities, which is distributed with the P6 program by Primavera, Inc., was used to verify the capabilities of SSO in terms of validity in dealing with a large real-world network. The schedule information (i.e., $i$, $D_i$, $r_i$, $e_i$, $m_i$, $f_i$, and $PS_i$, etc.) were entered into the system. The case study site was divided into 15 specific work areas. The occupation density functions for

each specific work area and the alternatives of execution pattern for each activity were acquired from plant construction experts.

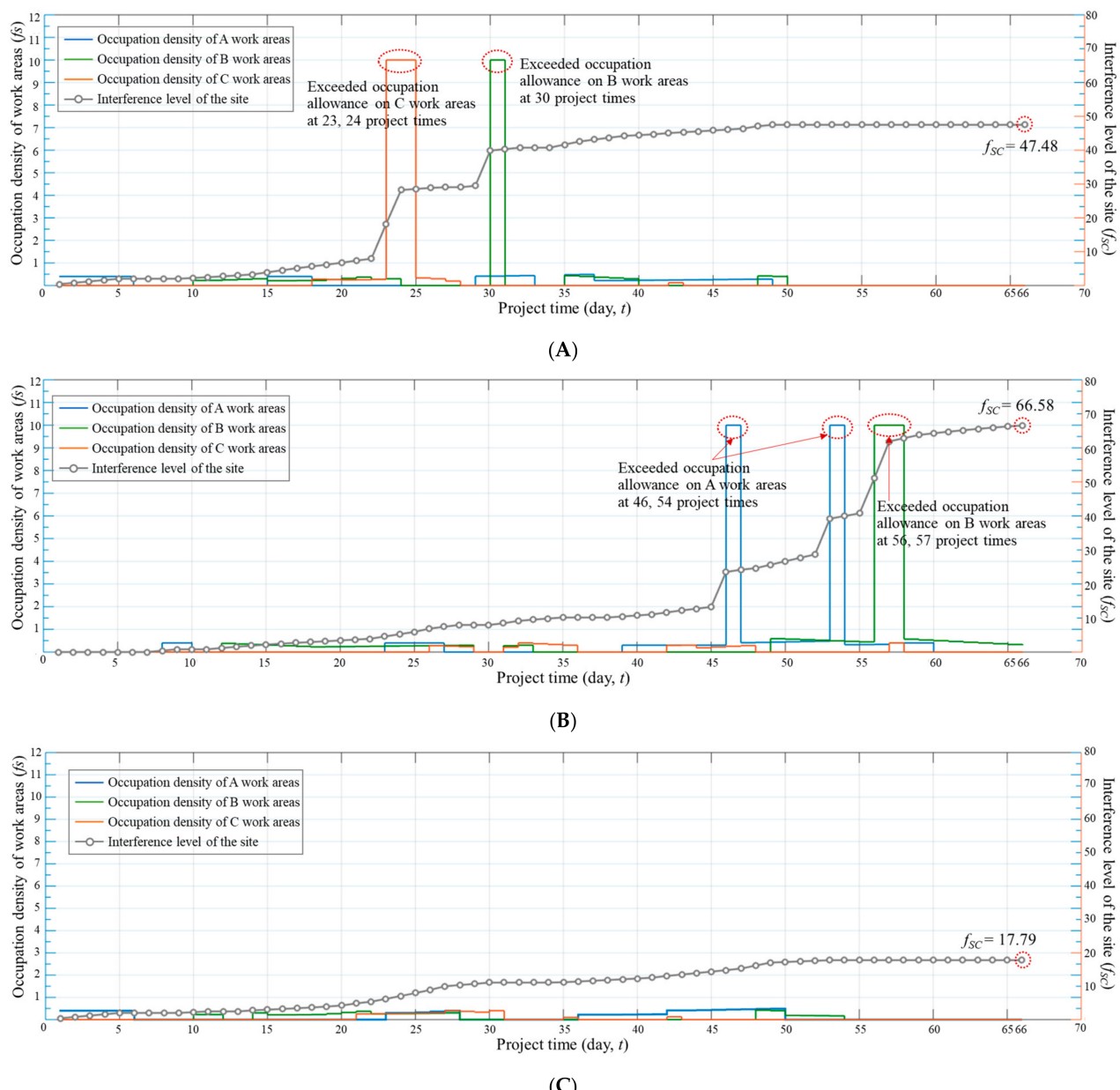

**Figure 8.** Occupation density of work areas and level of space interference obtained original schedule and SSO method. (**A**) Original schedule (based EST). (**B**) Original schedule (based LST). (**C**) After optimizing schedule using SSO.

When the case network was executed with EST and LST, the allowed occupation density of specific work areas was exceeded 12 and 8 times, respectively. Without space-constrained scheduling, project completion time and cost would inevitably increase. SSO identified the optimal set of pairs of execution patterns alternatives and deferred start times of activities that did not exceed the allowed occupation density of each specific work area. Therefore, the effect of stacking of trades on the work areas was minimized. Convergence time was only 325.8 s. The space interference level ($f_{SC}$) was reduced by 328% compared to the EST schedule. The schedule input data and output tables are not included due to a lack of space. However, there is no doubt that SSO finds near-global solutions even if the size of the network along with the number of activities and number of relationships is increased.

It should be noted that the time for defining the occupation density function increases as the number of specific work areas increases. Furthermore, the occupation density function should be defined by construction experts in the relevant field. However, with the rapid development of image processing technology, the technical basis for automatically generating occupation density functions is being prepared. Many studies on image processing-based schedule progress management [24–26] were conducted, and these studies performed various analyses regarding object detection, location tracking, motion detection, volume estimation, etc., to calculate rates of progress. In addition, the accuracy of image processing is becoming more sophisticated. It is possible to develop a system that executes image-processing analysis, acquire historical data regarding activities, and input it into Equation (3) to generate occupancy density functions automatically. In future studies, it is recommended to develop an automated schedule system incorporating SSO and image processing-based progress management methods.

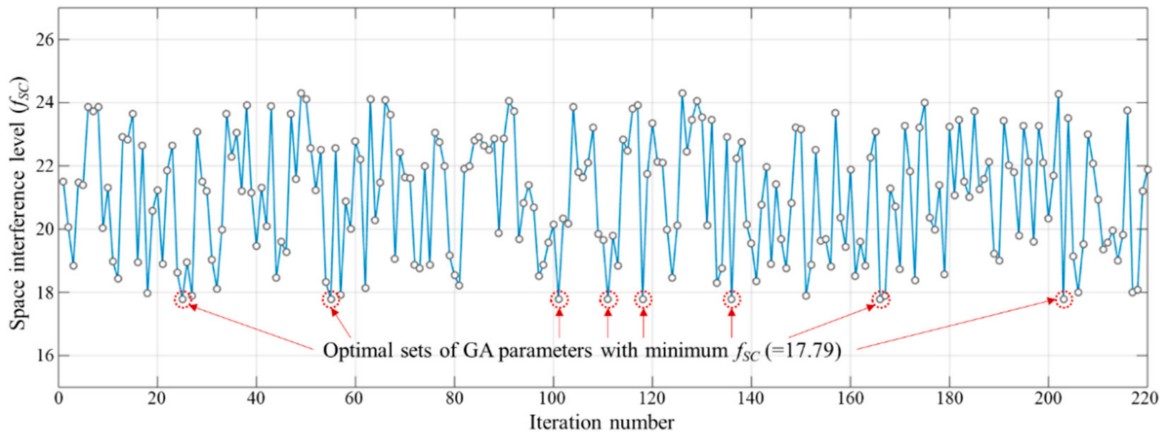

**Figure 9.** Results of sensitivity analysis.

## 5. Conclusions

This study proposes SSO, which finds the optimal space-constrained schedule that minimizes workspace conflicts and the effects of stacking of trades. First, SSO imports schedule information from the project database (e.g., Primavera P6), extracts IFC files of construction site area from the BIM model, and saves the data into the system memory. Then, the user classifies the entire site into specific work areas by considering the characteristics of the workspace and defines the occupation density function of each activity according to the attributes of each activity, allocated resources, and the rate of progress. Furthermore, SSO creates alternatives for the occupation density function (fs) obtained by changing the execution pattern of activity and defines the occupation density functions of the alternatives. SSO identifies an optimal space-constrained schedule using a GA optimization analysis technique. This method was developed as a MATLAB-based software, and users can operate the system easily by inputting the aforementioned information (i.e., P6 and BIM model interlinkage, specific work areas, occupation density function, and GA parameters). Furthermore, this system provides the user with visual information, such as (1) a set of alternatives for activity execution patterns, (2) a set of deferred start times of activities, (3) level of space interference, and (4) times when space interference occurred. This study provided, in detail, the computation process of calculating space interference levels, which change according to the combinations of alternatives for activity execution patterns and deferred start times of activities. According to the results of the case study, the near-global optimal solution that was identified by SSO resolved workspace conflicts, which are associated with construction delays and cost increases, and it reduced the space interference level ($f_{SC}$) by 328% compared to the EST schedule. Furthermore, SSO was validated for a large network. The effective time to execute the SSO method is after the

framework construction is completed. This is because, after the framework is completed, various types of work, such as electrical and mechanical work, etc., are carried out at the same time in a narrow workspace, whereby interference problems occur frequently.

This study utilizes (1) changing activity execution patterns and (2) deferring the start time of activities as a strategy for resolving workspace interference. However, the activity splitting approach is also being used in schedule management methods. Activity splitting is a resource allocation planning process that splits an activity into segments with re-allocated resources. Therefore, noncritical activities can be split to resolve the problem of space interference. In future studies, it is recommended to develop a methodology that incorporates activity splitting to find optimal space-constrained schedules. In addition, as mentioned in the case study, it is recommended that a follow-up study should be conducted to integrate image-processing-based progress management methods to develop an automated system that reduces user intervention in SSO. This study dealt with the space-constrained scheduling problem in a deterministic environment. However, since construction projects inevitably have duration and cost fluctuations, further research is needed on space-constrained scheduling optimization dealing with stochastic activity durations.

**Author Contributions:** Conceptualization, methodology, and software, H.-S.G.; validation, H.-S.G. and W.-S.S.; formal analysis and investigation, H.-S.G., W.-S.S. and Y.-J.P.; writing—original draft preparation, H.-S.G.; writing—review and editing, Y.-J.P.; visualization, W.-S.S. and Y.-J.P.; funding acquisition, H.-S.G. All authors have read and agreed to the published version of the manuscript.

**Funding:** This work is supported by the National Research Foundation of Korea (NRF) grant funded by the Korean government (MSIT) (No. 2021R1F1A1051109).

**Institutional Review Board Statement:** Not applicable.

**Informed Consent Statement:** Not applicable.

**Conflicts of Interest:** The authors declare no conflict of interest.

## Abbreviations

The following symbols are used in this article:

| | |
|---|---|
| $A$ | The set of alternatives of activity execution pattern |
| $a_{t,h}$ | The binary number that determines whether work area $h$ occupied simultaneously by multiple activities at time $t$ |
| $CR$ | The crossover rate |
| $D_i$ | The duration of activity $i$ |
| $e_{i,h}(p)$ | The work area occupied by equipment in work area $h$ when the rate of progress for activity $i$ is $p$ |
| $e_i$ | The number of equipment per day for activity $i$ |
| $es_i$ | The early start time of activity $i$ |
| $ef_i$ | The early finish time of activity $i$ |
| $f_i$ | The required temporary facilities for activity $i$ |
| $fs_{i,h,k}$ | The occupation density function for work area $h$ when the execution pattern alternative for activity $i$ is $k$ |
| $f_{i,h}(p)$ | The area occupied by temporary facilities in work area $h$ when the rate of progress for activity $i$ is $p$ |
| $f_{SC}(A, ST)$ | The objective function that computes the space interference level given a set of alternatives of activity execution pattern ($A$) and set of deferred start times ($ST$) |
| $g$ | The number of evolved generation that was necessary for reaching an optimal solution |
| $h$ | The index of a work area |
| $i$ | The index of an activity |
| $j$ | The index of a noncritical activity |
| $K_i$ | The number of alternatives for the occupation density function of activity $i$ |
| $k_i$ | The index of the alternatives of execution patterns for activity $i$ |

| | |
|---|---|
| $ls_i$ | The late start times of activity $i$ |
| $lf_i$ | The late finish times of activity $i$ |
| $m_i$ | The required amount of material for activity $i$ |
| $m_{i,h}(p)$ | The work area occupied by material in work area $h$ when the rate of progress for activity $i$ is $p$ |
| $MR$ | The mutation rate |
| $N_D$ | The matrix that stores the schedule information (i.e., activity index, activity duration, the number of laborers, the number of equipment, required amount of material, required temporary facilities, predecessor). |
| $N_{CP}$ | The set of noncritical activities |
| $PS_i$ | The predecessor of activity $i$ |
| $P$ | The rate of progress |
| $PS$ | The population size |
| $p_{i,t}$ | The progress rate of activity $i$ at time $t$ |
| $P_{t,h}$ | The penalty when the sum of the occupation density in work area $h$ at time $t$ is greater than 1 |
| $r_{i,h}(p)$ | The work area occupied by the labor in the work area $h$ when the rate of progress for activity $i$ is $p$ |
| $r_i$ | The number of labors per day for activity $i$ |
| $s_h$ | The area size of work area $h$ |
| $st_i$ | The start time of noncritical activity $i$ |
| $sd_i$ | The number of deferred days of start time for activity $i$ |
| $S_S$ | The search range of the optimal solution |
| $ST$ | The set of deferred start times |
| $S_D$ | The matrix that stores the vertex coordinates set and z value of the specific work area $h$ |
| $tf_i$ | The total floats of activity $i$ |

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
