# Peer review of "Space-Constrained Scheduling Optimization Method for Minimizing the Effects of Stacking of Trades"

_applsci, doi:10.3390/app112211047_

Round 1
Reviewer 1 Report
Dear authors,
your study is interesting and provides a novelty in the field of applying the heuristic optimization approach for solving construction site space interference and work productivity. Even though your paper is well-written, the model seems far too general. Hence, I suggest that you be more precise with the model, its sensitivity and connect it more closely to the verification case study. In this perspective, here are some of my suggestions for the major revision:
- the first sentence of the abstract seems like the middle of the chapter where the previous part is missing. Please revise your abstract as an independent but complete summary of your paper.
- the density function you present in figure 1 and refer to is presenting a case for electrical works and it is of great importance that you explain how does this correspond to your study.
- "trades" is not a common construction site term and can present a wide range of construction materials, elements, etc. You shoulčd be more precise, especially if you are dealing with bulk materials or discrete elements (i.e. products). Also, workspace and work areas are too general and they should correspond with the material or works you are dealing with.
- the literature review is weak. You should provide a more detailed literature review since you are dealing with a problem that is currently what we could call a mainstream. I can suggest an in-depth review paper that will help you find and filter relevant references for your study (i.e. Venkrbec et. al. 2018.: Construction process optimisation–review of methods, tools and applications. Građevinar, 70(07), pp. 593-606.).
- lines 153-173, this is a case of a hanging paragraph. You should either add this part below the 3.1. subchapter title or create a new one.
- in chapter 3., you should provide an explicit mathematical model of the problem. Stating the variables, their domains, and their nature, constraints, objective function.
- in chapter 4. you should provide a post-optimization analysis.
Kind regards
Author Response
The authors made extensive modifications in the paper based on the reviewers’ comments and thanks for constructive and detailed reviews.

Reviewer 2 Report
The work follows a timely research line that develops synergies between two research areas, GA and BIM, to address old construction productivity problems. In fact, while a factory is fine tunned along years, a construction site is a one-time event, and its own geometry changes along with the construction. It is a factory that is making itself along the way.
The text style is correct, and no severe flaws were found. The methodology is robust, and well explained, as are the results. The weakest point of the paper is that it does not address in a realistic, or even tentative way, the application of the research to real-world cases. This could be done in three places in the paper: in 3.1, the generation of the large amount of activity data that is needed (is it already available?); in 3.2 how that data can be linked with the BIM model or extracted from a 4D model, and what are the modelling requisits; in 4.2 not only the size of the network should be explored, but also a real world example; also in 4.2 what is the run time of the GA for a large scale calculation; in 5, is the direct application of the method still far away, how can it be integrated into the construction companies' workflow, is there a plan to use the developed method in a real worl example, can it be applied today, or what are the necessary steps for that to happen; in 5 how can the method be applied in a worksite where spaces are fluid, as when walls are being erected in the early stage of the construction; also in 5 how can the method be adpated to deal with uncertainty, namelly, statistical variation of the activities durations.
A number of improvements are detailed below, some of them are mere suggestions to make the text more fluid, others are of mandatory correction.
line 18 - GA is not yet defined
line 31 - "induce" or a similar word instead of "leads"? Also "affect decline" does not sound well
line 46 - "This study defines such the method as..." does not sound well
line 55 - the 2 in m2 should be superscript
line 57 to 58 - if work area density is sqm/worker when we have more room per worker the productivity should increase?
line 78 - "distinctive" is not well used here
line 82 - P6 has not been defined yet
line 102 - delete the "Next,", start the phrase with "The approaches..."
line 128 - strategies instead of methods?
line 160 - "area may have BEEN occupied"?
line 170 - in the form of A/THE software"
line 175 - "IN step 1"
line 176 and 341 - labors means laborers/workers?
Equation 1 (and others) - define the indexes
line 183 - explain why you use IfcProduct and provide a reference for the buildingSmart dictionary or similar that explains what it is
line 188 - "expertISE"
line 201 to 202 - is this automatic? How is this performed?
line 226 - rephrase the start of the sentence
line 215 to 219 - how are the areas for labor, equipment, etc. determined?
line 226 to 234 - text is hard to follow
line 280 - what is the MATBLAB algorithm? Which Toolbox? Provide a reference.
line 353 - the first ( is misplaced
line 372 - avoid words which are vague such as "many", be more precise, even if it is only an order of magnitude: 10, 100, 1000
Figure 8 - the vertical axis scales values are very different, maybe use a log scale, at least on the left?
line 402 - 328% precision is sufficient
line 405 - avoid words that are vague such as "tremendously"
line 424 - "SSO classifies": is this automatic?
Author Response

(The authors gave the same response as above.)

Reviewer 3 Report
This study is about value to practitioners because SSO provides an easy-to-use computerized tool that reduces the lengthy computations relative to data processing, and GAs. Test cases verify the validity of the computational method. This theme is very interesting and novel.
I recommend this article for publishing.
Author Response
Thanks for the good evaluation. We will improve the completeness of the article by reflecting the opinions of other reviewers.
Reviewer 4 Report
There are 18 references in the text. Of these, about 28% are less than 5 years old, and about 39% are more than 10 years old. The average date is 2012, which is acceptable.
I found the paper very interesting, and the proposed approach is very well detailed and schematized. Regarding the use of Genetic Algorithms, it was a smart choice, and it is well explained, namely the adopted chromosome is indicated in Figure 4, which is something important.
However, in my opinion the authors could have been a little more exhaustive in their introduction, namely by presenting other alternative approaches that are proposed by other authors, such as Particle Swarm Optimization.
The authors do not present to the reader the meaning of several acronyms, the first time they are presented in the text, such as BIM and GA in line 82, SA in line 135, TS in line 135, EST in line 360, LST in line 363, and so one.
Another issue that I was able to find is regarding the inconsistency how figures are referred in the text. Figures 1, 2 and 8 are referred as Fig., and all the others as Figure.
I believe that the paper is interesting for publishing, but only after a revision.
Author Response

(The authors gave the same response as above.)

Round 2
Reviewer 1 Report
Dear authors,
thank you for acknowledging my suggestions in the revised version of the paper. Perhaps the only remark I have for this version of the paper is to align the L activity with the activities M and K since the L activity is critical, in the network diagram in figure 6. Also, it would not hurt to add fictive finish-to-finish connections of activities M and K with activity L due to the mathematical connections that prove that M and K have no time slack (i.e. to close the graph). This I find important educational wise because keep in mind that students read such articles.
Overall, the paper seems better, and as far as I'm concerned can move forward in the publishing process. Good luck with your further work.
Kind regards
Author Response
The authors made modifications in the paper based on the reviewers’ comments and thanks for constructive and detailed reviews.

Reviewer 4 Report
The authors have addressed all my previous concerns, so I recommend publishing the paper.